# The Price of Robustness: Stable Classifiers Need Overparameterization

Jonas von Berg[*,1,2], Adalbert Fono[1,2], Massimiliano Datres[1,2], Sohir Maskey[1,2,3], and Gitta Kutyniok[1,2,4,5]

[1]Ludwig-Maximilians-Universität München
[2] Munich Center for Machine Learning (MCML)
[3]Aleph Alpha Research
[4]University of Tromsø
[5]DLR-German Aerospace Center
{berg, fono, datres, kutyniok}@math.lmu.de    sohir.maskey@aleph-alpha-research.com

## Abstract

The link between overparameterization, robustness, and generalization in discontinuous classifiers remains unclear. We establish generalization bounds that tighten with *class stability* – the expected distance to the decision boundary – yielding a *law of robustness for classification* that extends prior smoothness based settings. As a consequence, any interpolating model with $p \approx n$ parameters is necessarily *unstable*, implying that robust generalization requires overparameterization. For infinite function classes, we obtain analogous results through a stronger robustness measure, the *normalized co-stability*, defined via output margins. Empirical results support our theory: stability grows with model size and aligns closely with test performance.

## 1 Introduction

The generalization behavior of overparameterized neural networks presents fundamental challenges to classical statistical learning theory. Traditional complexity measures, such as parameter counts or spectral norms of weights, form the basis of many generalization bounds, including those derived from VC dimension theory [1] and Rademacher complexity [2]. However, these approaches do not adequately explain several empirical phenomena, e.g., *double descent* [3] and *benign overfitting* [4], where test performance improves beyond the interpolation threshold. Empirical studies further show that norm-based metrics often correlate poorly with generalization [5], while the *margin* – the distance to the decision boundary – emerges as a reliable predictor [6–8]. This suggests that generalization is governed not by microscopic weight norms but by *macroscopic simplicity*, the stability of predictions under perturbations. The *law of robustness* of Bubeck and Sellke [9] establishes a formal link between robustness, generalization, and overparameterization: smooth-

ness and overparameterization need to balance in order to ensure good generalization while overfitting. However, its reliance on smoothness assumptions excludes discontinuous classifiers. We address this limitation by introducing *class stability* and *normalized co-stability* – geometric, macroscopic measures of functional simplicity that extend robustness laws to classification.

## 2 Setup

We study binary classification on $(\mathcal{X} \times \{-1, 1\}, \mu)$, where $\mathcal{X} \subset \mathbb{R}^d$ is bounded and $\mathcal{F} \subset \{f : \mathcal{X} \to \{-1, 1\}\}$ a hypothesis class. Given $n$ i.i.d. samples $(x_i, y_i) \sim \mu$, the goal is to find $f \in \mathcal{F}$ minimizing a bounded loss $\ell$. We focus on the binary case; multi-class extensions follow by one-vs-all reduction ( A.4). A canonical loss is $\ell_{0\text{-}1}(y, y') = \mathbb{1}_{y \neq y'}$.

**Class stability.** Following Liu and Hansen [10], we measure robustness by the expected distance to the decision boundary. For $f : \mathcal{X} \to \{-1, 1\}$, define the *signed distance function*

$$d_f(x) = \begin{cases} d(x, f^{-1}(\{-1\})), & f(x) = 1, \\ -d(x, f^{-1}(\{1\})), & f(x) = -1, \end{cases} \quad (1)$$

where $d(x, A) = \inf_{y \in A} \|x - y\|_2$. The (unsigned) *margin* and the *class stability* are

$$h_f(x) = |d_f(x)|, \qquad S(f) = \mathbb{E}[h_f]. \quad (2)$$

Here $S(f)$ quantifies the average distance of samples to the decision boundary – a notion of macroscopic robustness. To extend our results to infinite, parameterized function classes, continuity in the parameterization is required. For this, we introduce a stronger, codomain-based notion of stability.

**Co-stability.** Any classifier can be written as $f = \text{sgn} \circ g$, where $g$ is Lipschitz continuous with constant $L(g)$ (see Lemma 2). This representation allows us

---

[*]Corresponding Author.

to define the (Lipschitz-normalized) *co-margin* and *co-stability* as

$$\bar{h}_g^*(x) = \frac{|g(x)|}{L(g)}, \qquad \bar{S}^*(g) = \mathbb{E}[\bar{h}_g^*(x)]. \qquad (3)$$

For the canonical choice $g = d_f$, they coincide with $S(f)$. Since perturbing $x$ by $r$ changes $g(x)$ by at most $L(g)r$, label flips require $r \geq |g(x)|/L(g)$, implying that in general we have the inequality

$$S(f) \geq \bar{S}^*(g). \qquad (4)$$

Normalized co-stability thus lower-bounds class stability and provides another scale-invariant robustness measure. Crucially, it guarantees that the score function $g$ remains, on average, at a nontrivial distance from the decision discontinuity, enabling extensions to infinite function classes.

**Isoperimetry.** To control how stable functions fit random labels, we assume $\mu$ satisfies concentration for Lipschitz functions:

$$\mathbb{P}(|f(x) - \mathbb{E}[f]| \geq t) \leq 2e^{-dt^2/(2cL^2)} \qquad (5)$$

for all bounded $L$-Lipschitz $f : \mathcal{X} \to \mathbb{R}$ and $t \geq 0$. This *c-isoperimetry* holds for Gaussian measures and for uniform measures on compact manifolds with positive curvature [9, 11]. Under the manifold hypothesis, $d$ represents intrinsic dimension.

# 3 Main Results: A Law of Robustness for Classification

We establish a *law of robustness for classification*, linking generalization to margin-based stability in discontinuous classifiers. In contrast to Lipschitz-based analyses [9], our bounds apply directly to discrete decision functions through the notions of *class stability* $S(f)$ and *normalized co-stability* $\bar{S}^*(g)$.

**Finite class Rademacher bound.** If the input distribution is $c$-isoperimetric and $\mathcal{F}$ is finite with $\min_{f \in \mathcal{F}} S(f) \geq S > 0$ and $\log |\mathcal{F}| \geq n$, then the Rademacher complexity satisfies

$$\mathcal{R}_{n,\mu}(\mathcal{F}) \lesssim \max\left\{ \frac{1}{\sqrt{n}}, \frac{1}{S}\sqrt{\frac{c \log |\mathcal{F}|}{n \, d}} \right\}. \qquad (6)$$

(Precise statement and proof: Theorem 4.)

**Infinite class Rademacher bound.** For parameterized classes $f = \text{sgn} \circ g_\omega$ with bounded parameter set $\mathcal{W} \subset \mathbb{R}^p$, where $g_\omega$ is Lipschitz in $\mathcal{X}$ ($L_\mathcal{X}(g) \leq L$) and Lipschitz in $\omega$, and $\bar{S}^*(g_\omega) \geq S^* > 0$, one obtains

$$\mathcal{R}_{n,\mu}(\mathcal{F}) \lesssim \max\left\{ \frac{1}{\sqrt{n}}, \frac{L}{S^*}\sqrt{\frac{c \, p}{n \, d}} \right\}. \qquad (7)$$

(Precise statement and proof: Theorem 5.)

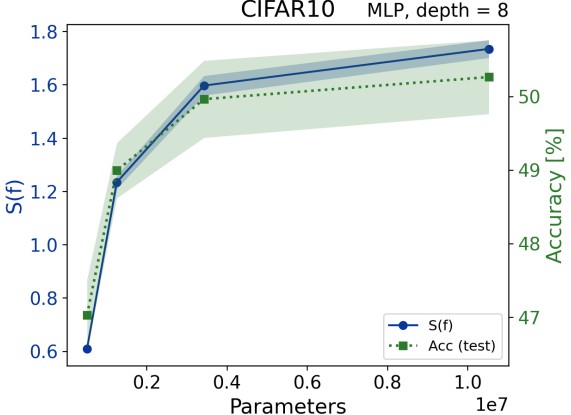

**Figure 1.** Class stability for MLPs trained on CIFAR-10.

**Law of Robustness.** Combining the above results with the standard generalization bound in terms of the Rademacher complexity [2] yields the following informal statement. If $\sigma^2 := \min_{f \in \mathcal{F}} R_{0-1}(f) > \varepsilon > 0$ and a classifier satisfies $\hat{R}_{0-1}(f) \leq \sigma^2 - \varepsilon$ for sufficiently large $n$, then with high probability

$$\frac{S^*(g)}{L(g)} \lesssim \frac{1}{\varepsilon}\sqrt{\frac{c \, p}{n \, d}}. \qquad (8)$$

Hence, simultaneously achieving low training error and high (co-)stability requires overparameterization on the order of $p \approx nd$. An analogous relation holds for finite function classes in terms of $S(f)$. (Precise finite and infinite formulations, together with proofs, are given in the appendix A.3.)

**Experiments.** We trained 4- and 8-layer MLPs of varying width on MNIST and CIFAR-10, estimating $S(f)$ via minimal $\ell_2$ adversarial radii and $\bar{S}^*$ via efficient lipschitz estimation (using the ECLIPSE method [12]). Both measures increase with width and correlate strongly with test accuracy. These trends, support our theory that (co-)stability, grows with overparameterization. Experimental details and further plots are provided in the appendix A.5.

# 4 Conclusion

Our results show that good generalization in overparameterized regimes hinges on sufficient stability. The inverse dependence on $S$ or $\bar{S}^*/L$ in our bounds indicates that stability reduces effective complexity, mitigating overfitting. In high dimensions, overparameterization becomes *necessary* for robust generalization: limited capacity forces a trade-off with (co-)stability, leading to large Lipschitz constants or low prediction confidence. This aligns with observations that large neural networks, including LLMs, generalize well despite overparameterization.

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

# A Appendix

## A.1 The Signed Distance Function

We collect the main properties of the signed distance function

$$d_f(x) := \begin{cases} d(x, f^{-1}(\{-1\})), & \text{if } f(x) = 1, \\ -d(x, f^{-1}(\{1\})), & \text{if } f(x) = -1, \end{cases} \quad (9)$$

where $d(x, A) := \inf_{y \in A} \|x - y\|_2$.

**Lemma 1.** *Let $\mathcal{X} \subset \mathbb{R}^d$ be bounded and path-connected, and let $f : \mathcal{X} \to \{-1, 1\}$. Then the signed distance function $d_f$ is 1-Lipschitz.*

This is a classical fact, a special case of the Eikonal equation. For completeness, we include a direct proof inspired by Liu and Hansen [10, Prop. 7.5].

*Proof.* **Case 1:** $f(x) = f(y)$. Assume w.l.o.g. $f(x) = f(y) = 1$. Let $(z_n)_n$ be a sequence in $f^{-1}(\{-1\})$ with $|d(y, z_n) - d_f(y)| \leq \frac{1}{n}$. Then

$$d_f(x) = d(x, f^{-1}(\{-1\})) \quad (10)$$
$$\leq d(x, z_n) \quad (11)$$
$$\leq \|x - y\|_2 + d(y, z_n) \quad (12)$$
$$\leq \|x - y\|_2 + d_f(y) + \frac{1}{n}. \quad (13)$$

Letting $n \to \infty$ and exploiting symmetry yields $|d_f(x) - d_f(y)| \leq \|x - y\|_2$.

**Case 2:** $f(x) \neq f(y)$. Assume w.l.o.g. $f(x) = 1$, $f(y) = -1$. Consider the line segment $L = \{(1 - t)x + ty : t \in [0, 1]\} \subset \mathcal{X}$ and define

$$w_1 = (1 - t_1)x + t_1 y, \quad (14)$$
$$t_1 := \inf\{t : f((1 - t)x + ty) = -1\}, \quad (15)$$
$$w_2 = (1 - t_2)x + t_2 y, \quad (16)$$
$$t_2 := \sup\{t : f((1 - t)x + ty) = 1\}. \quad (17)$$

Path-connectedness ensures $t_1 \leq t_2$, since otherwise the midpoint between $w_1$ and $w_2$ would be labeled both 1 and $-1$, a contradiction.

Thus,

$$|d_f(x) - d_f(y)| = d(x, f^{-1}(\{-1\})) + d(y, f^{-1}(\{1\})) \quad (18)$$
$$\leq \|x - w_1\|_2 + \|y - w_2\|_2 \quad (19)$$
$$\leq \|x - y\|_2. \quad (20)$$

□

**Lemma 2.** *Let $\mathcal{X} \subset \mathbb{R}^d$ and $f : \mathcal{X} \to \{-1, 1\}$ with $f^{-1}(\{1\})$ closed. Then $f$ can be represented as*

$$f(x) = \text{sgn}(d_f(x)), \quad (21)$$

*where we adopt the convention $\text{sgn}(0) = 1$.*

*Proof.* If $d_f(x) \neq 0$, the claim follows directly from the definition of $d_f$. If $d_f(x) = 0$, then $x \in f^{-1}(\{1\})$ by closedness, so $f(x) = 1 = \text{sgn}(0)$. □

**Remark 3.** *Lemma 2 justifies the representation $f = \text{sgn} \circ d_f$ used in the proof of Theorem 4. This link between classifiers and their signed distance functions is what allows stability arguments to be combined with smoothness-based tools.*

## A.2 Proofs of the Rademacher Bounds

We now provide proofs for the Rademacher bounds for finite and infinite function classes.

**Finite Rademacher Bound.** We begin by re-stating the assumptions.

(H1) $(\mathcal{X}, \mu)$ is a probability space with bounded sample space $\mathcal{X}$ and c-isoperimetric measure $\mu$;

(H2) the considered hypothesis class $\mathcal{F}$ of classifiers $f : \mathcal{X} \to \{-1, 1\}$ is finite, that is $|\mathcal{F}| < \infty$.

**Theorem 4** (Finite Rademacher Bound). *Suppose Assumptions (H1) and (H2) hold, and that $\min_{f \in \mathcal{F}} S(f) > S > 0$ with $\log |\mathcal{F}| \geq n$. Let us furthermore assume that $f^{-1}(\{1\})$ is closed and $\mathcal{X}$ path connected, then the empirical Rademacher complexity satisfies*

$$\mathcal{R}_{n,\mu}(\mathcal{F}) \leq K_2 \max \left\{ \frac{1}{\sqrt{n}}, \ \frac{\sqrt{c}}{S} \sqrt{\frac{\log |\mathcal{F}|}{nd}}, \right.$$
$$\left. 2 \exp\left(-\frac{dS^2}{8c}\right) \right\}. \quad (22)$$

*for an absolute constant $K_2 > 0$.*

*Proof.* By Lemma 2, every $f$ admits the representation $f = \mathrm{sgn} \circ d_f$. This allows us to follow the infinite-class analysis (see the proof of Theorem 5) without requiring the $\varepsilon$-net construction in Equation 30. By Lemma 1, the signed distance function $d_f$ is 1-Lipschitz, i.e., $L(d_f) = 1$ under the stated conditions. Moreover, recalling the definition of co-stability, we obtain

$$S^*(d_f) = \mathbb{E}[|d_f|] = \mathbb{E}[h_f] = S(f). \quad (23)$$

Plugging this into the general bound in Equation 25 gives the result. $\square$

**Infinite Rademacher bound** We extend the finite-class result to infinite function classes via a covering-number argument, for which the Lipschitz continuity of the parameterization plays a crucial role. To this end, we introduce a new regularity assumption that replaces the finiteness condition (H2).

(H3) The hypothesis class $\mathcal{F}$ is of the form $\mathcal{F} = \mathrm{sgn} \circ \mathcal{G}$, where $\mathcal{G} = \{g_w : \mathcal{X} \to [-1, 1] : w \in \mathcal{W}\}$ is a parameterized class of Lipschitz continuous functions. The parameter space $\mathcal{W} \subset \mathbb{R}^p$ is bounded with $\mathrm{diam}(\mathcal{W}) \leq W$, and the parameterization is Lipschitz continuous, i.e.,

$$\|g_{w_1} - g_{w_2}\|_\infty \ \leq \ J \|w_1 - w_2\|. \quad (24)$$

**Theorem 5** (Infinite Rademacher Bound). *Under assumptions (H1) and (H3), suppose that $S^*(g) > S^* > 0$ and $L(g) \leq L$ for all $g \in \mathcal{G}$. Furthermore, assume that $p \geq n$. Then, for any covering precision $\tilde{\varepsilon} > 0$,*

$$\mathcal{R}_{n,\mu}(\mathcal{F})$$
$$\leq K \max \left\{ \sqrt{\frac{1}{n}}, \ \frac{L}{S^*} \sqrt{\frac{cp}{nd}} \sqrt{\log(1 + 60WJ\tilde{\varepsilon}^{-1})}, \right.$$
$$\left. 2 \exp\left(-\frac{dS^{*2}}{8cL^2}\right), \ \frac{J}{S^*}\tilde{\varepsilon} \right\}. \quad (25)$$

*where $K > 0$ is an absolute constant independent of $p, n, d, S^*, c, L, J, \tilde{\varepsilon}, W$.*

*Proof.* Given any discontinuous classifier $f_w = \mathrm{sgn} \circ g_w$ for $g_w \in \mathcal{G}$, define its Lipschitz continuous approximation for $\gamma > 0$ as

$$F_{f_w} = \mathrm{sgn}_\gamma \circ g_w, \quad (26)$$

where

$$\mathrm{sgn}_\gamma(t) := \begin{cases} -1, & t \leq -\gamma, \\ \frac{t}{\gamma}, & t \in [-\gamma, \gamma], \\ 1, & t \geq \gamma. \end{cases} \quad (27)$$

This approximation satisfies the useful property that both $F_{f_w}$ and the absolute difference $|f_w - F_{f_w}|$ are Lipschitz continuous in both the input space $\mathcal{X}$ and the weight space $\mathcal{W}$, with

$$L(|\mathrm{sgn}_\gamma \circ g_w - \mathrm{sgn} \circ g_w|) = L(\mathrm{sgn}_\gamma \circ g_w) = \frac{L(g_w)}{\gamma}. \quad (28)$$

Using Lipschitz-continuous surrogates $F_f$, we decompose the Rademacher complexity into a smooth component, to which the analysis of Bubeck and Sellke [9] applies, and a residual term.

$$\mathcal{R}_{n,\mu}(\mathcal{F}) = \frac{1}{n}\mathbb{E}_{\sigma,x}\left[\sup_{f \in \mathcal{F}} \left|\sum_{i=1}^n \sigma_i f(x_i)\right|\right]$$

$$\leq \frac{1}{n}\mathbb{E}_{\sigma,x}\left[\sup_{f \in \mathcal{F}} \left|\sum_{i=1}^n \sigma_i F_f(x_i)\right|\right]$$

$$+ \frac{1}{n}\mathbb{E}_{\sigma,x}\left[\sup_{f \in \mathcal{F}} \left|\sum_{i=1}^n \sigma_i (f - F_f)(x_i)\right|\right]$$

$$\leq C_1 \frac{1}{\sqrt{n}} + C_2 \frac{L}{\gamma} \sqrt{\frac{cp}{nd}} \sqrt{\log(1 + 60WJ/\tilde{\varepsilon})}$$

$$+ \frac{1}{n}\mathbb{E}_{\sigma,x}\left[\sup_{f \in \mathcal{F}} \left|\sum_{i=1}^n \sigma_i (f - F_f)(x_i)\right|\right]. \quad (29)$$

Here the parameter $\tilde{\varepsilon} > 0$ is related to a $\tilde{\varepsilon}$-net of $\mathcal{W}$, which we denote by $\mathcal{W}_{\tilde{\varepsilon}}$. Note, that $|\mathcal{W}_{\tilde{\varepsilon}}| \leq (1 + 60WJ\tilde{\varepsilon}^{-1})^p$ (see e.g. [11] Corollary 4.2.13) so the same holds true for the induced net $\mathcal{F}_{\tilde{\varepsilon}} = \{\mathrm{sgn} \circ g_w : w \in \mathcal{W}_{\tilde{\varepsilon}}\}$, which allows us to treat the remaining expectation by subdividing the supremum:

$$\frac{1}{n}\,\mathbb{E}_{\sigma,x}\Big[\sup_{f\in\mathcal{F}}\Big|\sum_{i=1}^{n}\sigma_i(f - F_f)(x_i)\Big|\Big]$$

$$= \frac{1}{n}\,\mathbb{E}_{\sigma,x}\Big[\sup_{\substack{w_{\tilde{\varepsilon}}\in\mathcal{W}_{\tilde{\varepsilon}}\\\|w-w_{\tilde{\varepsilon}}\|\le\tilde{\varepsilon}}}\Big|\sum_{i=1}^{n}\sigma_i\big(f_w - F_{f_w}\big)(x_i)\Big|\Big]$$

$$\le \frac{1}{n}\,\mathbb{E}_{x}\Big[\sup_{w_{\tilde{\varepsilon}}\in\mathcal{W}_{\tilde{\varepsilon}}}\sum_{i=1}^{n}|f_{w_{\tilde{\varepsilon}}} - F_{f_{w_{\tilde{\varepsilon}}}}|(x_i)\Big]$$

$$+ \frac{1}{n}\,\mathbb{E}_{x}\Big[\sup_{\substack{w_{\tilde{\varepsilon}}\in\mathcal{W}_{\tilde{\varepsilon}}\\\|w-w_{\tilde{\varepsilon}}\|\le\tilde{\varepsilon}}}\sum_{i=1}^{n}\big|\,|f_w - F_{f_w}|$$

$$-\,|f_{w_{\tilde{\varepsilon}}} - F_{f_{w_{\tilde{\varepsilon}}}}|\,\big|(x_i)\Big].\quad(30)$$

By Lipschitz continuity of the parameterization and of $|f - F_f|$ (Equation 28), we obtain

$$\|\,|f_w - F_{f_w}| - |f_{w_{\tilde{\varepsilon}}} - F_{f_{w_{\tilde{\varepsilon}}}}|\,\|_\infty \le \frac{J}{\gamma}\,\tilde{\varepsilon}\qquad(31)$$

for any $w_{\tilde{\varepsilon}}\in\mathcal{W}_{\tilde{\varepsilon}}$ and $w\in B_{\tilde{\varepsilon}}(w_{\tilde{\varepsilon}})$, so that

$$\frac{1}{n}\,\mathbb{E}_{x}\Big[\sup_{w_{\tilde{\varepsilon}}\in\mathcal{W}_{\tilde{\varepsilon}}}\sum_{i=1}^{n}\big|\,|f_w - F_{f_w}|(x_i)$$

$$-\,|f_{w_{\tilde{\varepsilon}}} - F_{f_{w_{\tilde{\varepsilon}}}}|(x_i)\,\big|\Big] \le \frac{J}{\gamma}\,\tilde{\varepsilon}.\quad(32)$$

Note, that the expectation of the maximum of $N$ subgaussians $X_1,\ldots,X_N$ with variance proxy $\sigma^2$ scales as

$$\mathbb{E}\Big[\max_{1\le i\le N}|X_i|\Big] \le \sigma\sqrt{2\log(2N)},\qquad(33)$$

see for instance [13]. The first expectation in Equation 30 can be bounded using Equation 33, since it corresponds – up to centering – to a maximum of sub-Gaussian random variables with variance proxy $\sigma^2 = \frac{L^2}{\gamma^2}\frac{cn}{d}$. Therefore,

$$\frac{1}{n}\,\mathbb{E}_{x}\Big[\sup_{w_{\tilde{\varepsilon}}\in\mathcal{W}_{\tilde{\varepsilon}}}\sum_{i=1}^{n}|f_{w_{\tilde{\varepsilon}}} - F_{f_{w_{\tilde{\varepsilon}}}}|(x_i)\Big] =$$

$$\frac{1}{n}\,\mathbb{E}_{x}\Big[\sup_{w_{\tilde{\varepsilon}}\in\mathcal{W}_{\tilde{\varepsilon}}}\sum_{i=1}^{n}|f_{w_{\tilde{\varepsilon}}} - F_{f_{w_{\tilde{\varepsilon}}}}|(x_i) - \mathbb{E}[|f_{w_{\tilde{\varepsilon}}} - F_{f_{w_{\tilde{\varepsilon}}}}|]\Big]$$

$$+ \sup_{w_{\tilde{\varepsilon}}\in\mathcal{W}_{\tilde{\varepsilon}}}\mathbb{E}[|f_{w_{\tilde{\varepsilon}}} - F_{f_{w_{\tilde{\varepsilon}}}}|]$$

$$\le C_3\frac{L}{\gamma}\sqrt{\frac{c}{nd}}\sqrt{p\log(1 + 60WJ\tilde{\varepsilon}^{-1})}$$

$$+ \sup_{w_{\tilde{\varepsilon}}\in\mathcal{W}_{\tilde{\varepsilon}}}\mathbb{E}_{x}[|f_{w_{\tilde{\varepsilon}}} - F_{f_{w_{\tilde{\varepsilon}}}}|].\quad(34)$$

Finally, for every $f\in\mathcal{F}$,

$$\mathbb{E}_{x}[|f - F_f|] = \int_{\mathcal{X}}|f(x) - F_f(x)|\,d\mu(x)$$

$$\le \mathbb{P}(g(x)\in[-\gamma,\gamma]).\quad(35)$$

Choosing $\gamma = \frac{S^*(g)}{2}$, we obtain by the definitions of co-margin, and once again isoperimetry (since the co-margin inherits the Lipschitzness from $g$ by design)

$$\mathbb{P}\left(g(x)\in[-\gamma,\gamma]\right) = \mathbb{P}\left(|g(x)|\le\frac{S^*(g)}{2}\right)$$

$$\le \mathbb{P}\left(|h_g^*(x) - S^*(g)|\ge\frac{S^*(g)}{2}\right)$$

$$\le 2\exp\left(-\frac{d\,S^*(g)^2}{8cL(g)^2}\right) \le 2\exp\left(-\frac{d\,S^{*2}}{8cL^2}\right)$$

$$= 2\exp\left(-\frac{d\,\bar{S}^{*2}}{8c}\right).\quad(36)$$

Putting it all together, we have

$$\mathcal{R}_{n,\mu}(\mathcal{F})$$

$$\le C_1\frac{1}{\sqrt{n}} + C_2'\frac{L}{S^*}\sqrt{\frac{c}{nd}}\sqrt{p\log(1 + 60WJ\tilde{\varepsilon}^{-1})} + \frac{2J}{S^*}\tilde{\varepsilon}$$

$$+ 2\exp\left(-\frac{d\,S^{*2}}{8cL^2}\right),\quad(37)$$

for absolute constants $C_1, C_2'$, independent of $p, n, d, S^*, c, L, J, \tilde{\varepsilon}, W$. $\qquad\square$

## A.3 Proof of the Law of Robustness

Next, we provide the proof of the law of robustness for classification problems.

**Corollary 6** (Law of Robustness for Discontinuous Functions). *Assume we are in the setting of Theorem 4. Let $p := \log|\mathcal{F}| \ge n$. Fix $\varepsilon,\delta\in(0,1)$ and consider the 0–1 loss $\ell_{0-1}$. There exists an absolute constant $K > 0$ such that, if*

1. *the minimal risk $\sigma^2 := \min_{f\in\mathcal{F}} R_{0-1}(f)$ satisfies $\sigma^2\ge\varepsilon$, and*

2. *the sample size $n$ is large enough to ensure (i) $\frac{K}{\sqrt{n}} < \frac{\varepsilon}{3}$ and (ii) $\sqrt{\frac{2\log(2/\delta)}{n}} < \frac{\varepsilon}{2}$,*

*then with probability at least $1 - \delta$ (over the sample), the following holds uniformly for all $f\in\mathcal{F}$:*

$$\hat{R}_{0-1}(f) \le \sigma^2 - \varepsilon \implies$$

$$S(f) < \max\left\{\frac{3K}{\varepsilon}\sqrt{\frac{c\log|\mathcal{F}|}{nd}},\ \sqrt{\frac{8c}{d}\log\left(\frac{6K}{\varepsilon}\right)}\right\}.$$
$$(38)$$

*Proof.* Let $K > 0$ be an absolute constant such that Equation 22 holds, and define the threshold stability

$$S_* = S_*(p, n, d, \varepsilon)$$

$$:= \max\left\{\frac{3K}{\varepsilon}\sqrt{\frac{c\log|\mathcal{F}|}{nd}},\ \sqrt{\frac{8c}{d}\log\left(\frac{6K}{\varepsilon}\right)}\right\}.$$
$$(39)$$

Then, Theorem 4, together with condition 2(i), implies that

$$\mathcal{R}_{n,\mu}(\mathcal{F}_{S_*})$$

$$\leq K \max\left\{ \frac{1}{\sqrt{n}}, \ \frac{\sqrt{c}}{S_*}\sqrt{\frac{\log|\mathcal{F}|}{nd}}, \ 2\exp\left(-\frac{dS_*^2}{8c}\right) \right\}$$

$$\leq \varepsilon/3, \quad (40)$$

where $\mathcal{F}_{S_*} := \{f \in \mathcal{F} : S(f) \geq S_*\}$ is the subset of functions in $\mathcal{F}$ with stability at least $S_*$. Hence, applying the standard generalization inequality in terms of the Rademacher complexity [2], together with condition 2(ii), yields that with probability at least $1 - \delta$:

$$\sup_{f \in \mathcal{F}_{S_*}} \left(R_{0\text{-}1}(f) - \hat{R}_{0\text{-}1}(f)\right)$$

$$\leq 2\mathcal{R}_{n,\mu}(\ell_{0\text{-}1} \circ \mathcal{F}_{S_*}) + \sqrt{\frac{2\log(2/\delta)}{n}}$$

$$\leq \mathcal{R}_{n,\mu}(\mathcal{F}_{S_*}) + \frac{\varepsilon}{2} < \varepsilon, \quad (41)$$

where we additionally used

$$\mathcal{R}_{n,\mu}(\ell_{0\text{-}1} \circ \mathcal{F}) \leq \frac{1}{2}\mathcal{R}_{n,\mu}(\mathcal{F}), \quad (42)$$

in the second step. In particular, we can bound the probability

$$\mathbb{P}(\forall f \in \mathcal{F}_{S_*} : \hat{R}_{0\text{-}1}(f) > \sigma^2 - \varepsilon)$$

$$\geq \mathbb{P}(\forall f \in \mathcal{F}_{S_*} : R_{0\text{-}1}(f) - \hat{R}_{0\text{-}1}(f) < \varepsilon) \geq 1 - \delta, \quad (43)$$

where the first inequality follows from

$$R_{0\text{-}1}(f) - \hat{R}_{0\text{-}1}(f) < \varepsilon \overset{\text{condition 1.}}{\Longrightarrow}$$

$$\sigma^2 - \hat{R}_{0\text{-}1}(f) < \varepsilon \implies \hat{R}_{0\text{-}1}(f) > \sigma^2 - \varepsilon. \quad (44)$$

Decomposing this probability into two disjoint events

$$1 - \delta \leq \mathbb{P}(\forall f \in \mathcal{F}_{S_*} : \hat{R}_{0\text{-}1}(f) > \sigma^2 - \varepsilon)$$

$$= \mathbb{P}(\forall f \in \mathcal{F} : \hat{R}_{0\text{-}1}(f) > \sigma^2 - \varepsilon)$$

$$+ \mathbb{P}(\exists f \in \mathcal{F}_{S_*}^c : \hat{R}_{0\text{-}1}(f) \leq \sigma^2 - \varepsilon), \quad (45)$$

enables us to easily recognize that the expression exactly characterizes the probability that the following implication, and thereby the result, holds uniformly for all $f \in \mathcal{F}$:

$$\hat{R}_{0\text{-}1}(f) \leq \sigma^2 - \varepsilon \implies S(f) < S_*. \quad (46)$$

Indeed, the implication above holds if, for a given data sample $(x_i, y_i)_{i=1}^n$, either

- no function $f \in \mathcal{F}$ satisfies $\hat{R}_{0\text{-}1}(f) \leq \sigma^2 - \varepsilon$, or

- any such $f$ lies in $\mathcal{F}_{S_*}^c$, that is, $S(f) < S_*$,

which is the case with probability at least $1 - \delta$ due to Equation 45. $\square$

With the same reasoning and Theorem 5, we obtain a law of robustness for infinite classes.

**Corollary 7** (Law of Robustness for Infinite Function Classes)**.** *Assume we are in the setting of Theorem 5, and fix $\varepsilon, \delta \in (0, 1)$. Consider the 0–1 loss $\ell_{0-1}$. There exists an absolute constant $K > 0$ such that, if*

1. *the minimal risk $\sigma^2 := \min_{f \in \mathcal{F}} R_{0-1}(f)$ satisfies $\sigma^2 \geq \varepsilon$, and*

2. *the sample size $n$ is large enough so that (i) $\frac{K}{\sqrt{n}} < \frac{\varepsilon}{3}$ and (ii) $\sqrt{\frac{2\log(2/\delta)}{n}} < \frac{\varepsilon}{2}$,*

*then with probability at least $1 - \delta$, for all $\tilde{\varepsilon} > 0$, the following holds uniformly for all $g \in \mathcal{G}$ and $f_g = sgn \circ g$:*

$$\hat{R}_{0-1}(f_g) \leq \sigma^2 - \varepsilon \implies$$

$$\frac{S^*(g)}{L(g)} < \max\left\{ \frac{3K}{\varepsilon}\sqrt{\frac{p}{nd}}\sqrt{c\log(1 + 60WJ\tilde{\varepsilon}^{-1})} \right.$$

$$\left. , \sqrt{\frac{8c}{d}\log\left(\frac{6K}{\varepsilon}\right)} \right\}. \quad (47)$$

## A.4 Multi-Class Classification

In this section, we briefly outline how our results extend to categorical distributions with $\mathcal{C} \in \mathbb{N}$ classes. We assume that a classifier is given by

$$f : \mathcal{X} \to \{0, 1\}^{\mathcal{C}}, \quad (48)$$

with exactly one non-zero entry for each $x \in \mathcal{X}$. The adaptations of the conditions in (H3) to the multi-class setting can be formalized as follows.

(H3)' The hypothesis class has the form $\mathcal{F} = \text{argmax} \circ \mathcal{G}$, where $\mathcal{G} = \{g_w : \mathcal{X} \to [0, 1]^{\mathcal{C}} : w \in \mathcal{W}\}$ is a parameterized family of Lipschitz functions. The parameter space $\mathcal{W} \subset \mathbb{R}^p$ is bounded with $\text{diam}(\mathcal{W}) \leq W$, and the parameterization is Lipschitz:

$$\|g_{w_1} - g_{w_2}\|_\infty \leq J\|w_1 - w_2\|. \quad (49)$$

Thus, we can interpret $g \in \mathcal{G}$ as representing the class probabilities.

**Remark 8.** *For binary classification, i.e. $\mathcal{C} = 2$, the classifiers are of the form $f : \mathcal{X} \to \{0, 1\}^2$, instead of $f : \mathcal{X} \to \{-1, 1\}$, as considered earlier. However, one can translate between these representations by post-composing with either*

$$\alpha(x_1, x_2) := x_1 - x_2 \quad or \quad \beta(x) := \left(\frac{x+1}{2}, \frac{1-x}{2}\right). \quad (50)$$

*By the contraction principle for Rademacher complexity, it is therefore sufficient to compute the complexity for one of these models.*

As in the binary case, our proofs start by considering the Rademacher complexity of the function class $\mathcal{F}$:

$$\mathcal{R}_{n,\mu}(\mathcal{F}) = \frac{1}{n}\,\mathbb{E}^{\sigma_{ij},x_i}\left[\sup_{f\in\mathcal{F}}\Big|\sum_{i=1}^{n}\sum_{j=1}^{\mathcal{C}}\sigma_{ij}f_j(x_i)\Big|\right] \tag{51}$$

$$\leq \sum_{j=1}^{\mathcal{C}}\frac{1}{n}\,\mathbb{E}^{\sigma_{ij},x_i}\left[\sup_{f\in\mathcal{F}}\Big|\sum_{i=1}^{n}\sigma_{ij}f_j(x_i)\Big|\right]. \tag{52}$$

Each summand corresponds to a binary classification problem with a one-vs-all classifier $f_j$. Indeed, $f_j$ is $\frac{2}{S(f)-t}$-Lipschitz on $A_t(f)$. Transforming via

$$f_j \mapsto 2f_j - 1 : \mathcal{X} \to \{-1,1\}, \tag{53}$$

we can follow the same reasoning as in A.2, obtaining, up to a linear factor of $\mathcal{C}$, the same result as the first part of Theorem 4, generalized to the multi-class setting.

Similarly, under assumption (H3), we can write

$$2f_j - 1 = \mathrm{sgn}\big(g_j - \max_{i\neq j} g_i(x)\big), \tag{54}$$

which allows us to proceed as in Theorem 5 to obtain a multi-class generalization of Theorem 5 and Corollary 7. The only minor difference lies in bounding the term in Equation 35:

$$\mathbb{E}[|f_j - F_{f_j}|] \leq \mathbb{P}\big[|g_j(x) - \max_{i\neq j} g_i(x)| \leq \gamma\big]. \tag{55}$$

Choosing $\gamma = \frac{S^*(g)}{2}$, we use that for all $j$, $|g_j(x) - \max_{i\neq j} g_i(x)| > h_g^*(x)$, which yields

$$\mathbb{P}\big[|g_j(x) - \max_{i\neq j} g_i(x)| \leq \tfrac{S^*(g)}{2}\big] \tag{56}$$

$$\leq \mathbb{P}\big[|h_g^*(x) - S^*(f)| \geq \tfrac{S^*(g)}{2}\big] \tag{57}$$

$$\leq 2\exp\Big(-\frac{d\,S^*(g)^2}{8cL(g)^2}\Big) \tag{58}$$

$$\leq 2\exp\Big(-\frac{d\,S^{*2}}{8cL^2}\Big) \tag{59}$$

$$= 2\exp\Big(-\frac{d\,\bar{S}^{*2}}{8c}\Big). \tag{60}$$

We conclude that all results extend naturally to the multi-class case. The main concepts are summarized below.

- **Isoperimetry**:

$$\mathbb{P}(\|f(x) - \mathbb{E}[f]\|_\infty \geq t) \leq 2\exp\Big(-\frac{dt^2}{2cL^2}\Big) \tag{61}$$

- **Rademacher complexity**

$$\mathcal{R}_{n,\mu}(\mathcal{F}) = \frac{1}{n}\mathbb{E}^{\sigma_{i,j},x_i}\left[\sup_{f\in\mathcal{F}}\Big|\sum_{i=1}^{n}\sum_{j=1}^{\mathcal{C}}\sigma_{ij}f_j(x_i)\Big|\right] \tag{62}$$

- **Margin**

$$h_f(x) = \sum_{j=1}^{\mathcal{C}} h_f^j(x) \tag{63}$$

$$h_f^j(x) := \inf\{\|x - z\|_2 : f(z) \neq j,\ z \in \mathbb{R}^d\} \tag{64}$$

- **Class stability**

$$S(f) = \sum_{j=1}^{\mathcal{C}} S(f)^j, \quad S(f)^j := \mathbb{E}[h_f^j] \tag{65}$$

- **Co-margin**

$$h_g^*(x) = \sum_{j=1}^{\mathcal{C}} h_g^{*j}(x) \tag{66}$$

$$h_g^{*j}(x) := \max\big(0, g_j(x) - \max_{i\neq j} g_i(x)\big) \tag{67}$$

- **Co-stability**

$$S^*(g) = \sum_{j=1}^{\mathcal{C}} S^{*j}(g), \quad S^{*j}(g) := \mathbb{E}[h_g^{*j}] \tag{68}$$

## A.5 Experimental Details for Stability Measurement

**Training setup.** To empirically validate our robustness law, we trained fully connected MLPs on MNIST and CIFAR-10 datasets. Each model has 4 hidden layers with widths $w \in \{128, 256, 512, 1024, 2048\}$ for MNIST and up to $w = 1024$ for CIFAR10. All models use ReLU activations, batch normalization, and were initialized with standard parametrization. Training was conducted using the Adam optimizer [14] for the embedding and output layers, and the Muon optimizer [15] for the hidden layers. Models were trained with a batch size of 256 and learning rate $10^{-3}$, until at least 99% training accuracy was achieved, ensuring (near) interpolation. We further used sharpness-aware optimization based on [16, 17] to reduce variance of the normalized co-stability on MNIST.

**Parameter counts and normalization.** For each model, we recorded the total number of trainable parameters $p$, input dimension $d$, and total number of training samples $n$.

**Stability estimation.** Class stability $S(f)$ was computed using adversarial perturbation analysis. We performed a suite of $\ell_2$-based attacks (FGSM, PGD, DeepFool, and L2PGD [18–20]) using the Foolbox library [21]. For each input $x$, we recorded the minimum perturbation norm required to change the classifier's prediction, over a grid of radii $\mathbf{r} = (0.002, 0.01, 0.05, 0.1)$. The final stability score $S(f)$ was taken as the average $\ell_2$ distance across the dataset.

**Normalized Co-Stability estimation.** The empirical co-stability $S^*(g)$ is computed via the multi-class margin

$$g_j(x) - \max_{i \neq j} g_i(x), \qquad j = \arg\max_i g_i(x), \quad (69)$$

averaged over the dataset. We estimate the Lipschitz constant $L(g)$ using the efficient ECLIPSE method [12], and report the normalized ratio $S^*(g)/L(g)$ as a function of model size.

**Implementation.** Training and evaluation code is implemented in PyTorch [22]. For MLPs, images were flattened to vectors. Attack evaluations were conducted over the full dataset (train and test).

**Reproducibility.** All experiments were run with multiple random seeds $\{0, 1, 2, 3, 4\}$, and mean with standard deviation are reported. Our code is available at anonymous GitHub.

**Results.** Figure A.1 shows that, for MLPs, both class stability $S(f)$ and normalized co-stability $S^*(g)/L(g)$ increase consistently with model size. The observed saturation of (normalized co-) stability aligns with theoretical intuition: the Bayes classifier admits a finite (normalized co-) stability level, and pushing beyond this level necessarily reduces accuracy – an instance of the robustness/accuracy trade-off extensively discussed in the literature [23–25]. Accordingly, we expect stability to plateau once models approach the Bayes decision boundary. For CIFAR-10, although test accuracy remains far below the Bayes optimal (around 50%), the same reasoning applies relative to the best classifier achievable within the restricted MLP architecture.

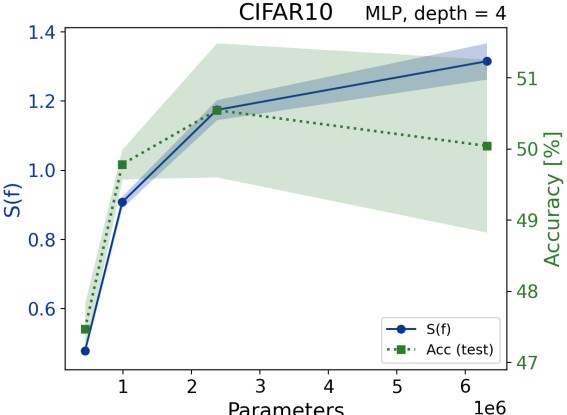

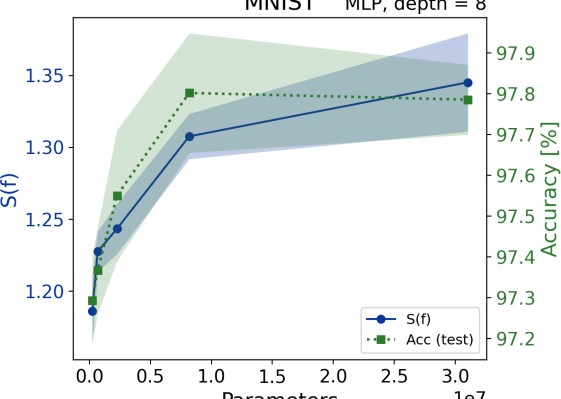

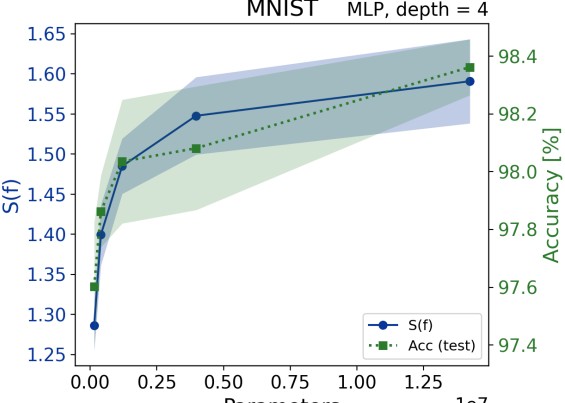

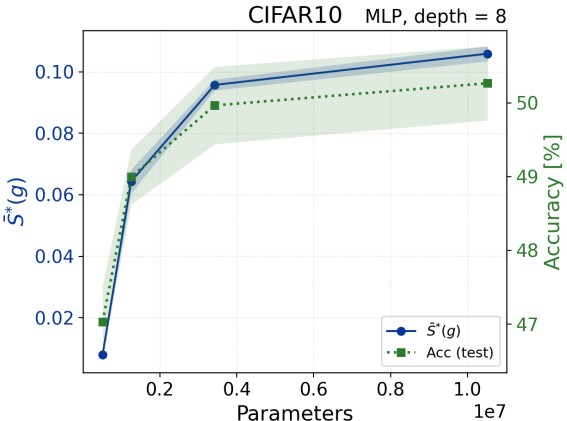

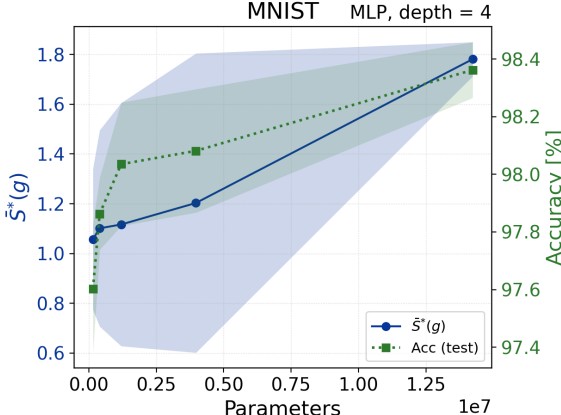

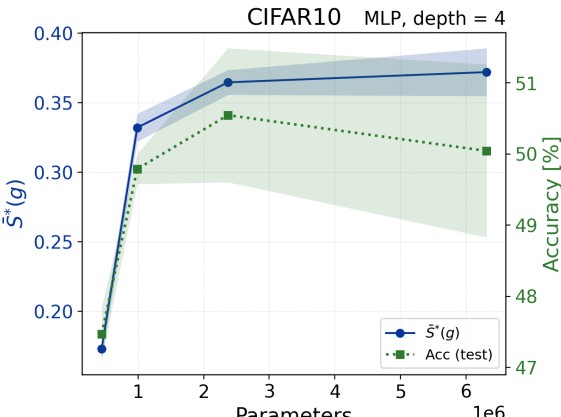

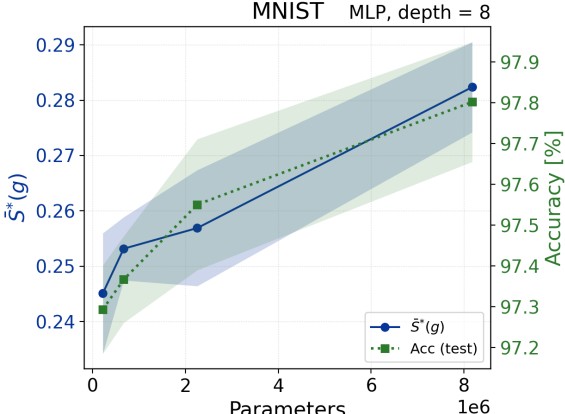

**Figure A.1.** Stability measures for 4- and 8-layer MLPs trained on MNIST and CIFAR-10.

