# OpenReview forum: "The Price of Robustness: Stable Classifiers Need Overparameterization"
_NLDL.org/2026/Abstracts_Track — NLDL 2026 Abstracts_

### Official Review · Reviewer_eZxJ · 2025-10-31

**Soundness:** 3
**Correctness:** 3
**Rating:** 5
**Confidence:** 2

**Summary:**

This paper extends known results on generalization and overparameterization from continuous to discontinuous classifiers. It shows that overparameterization improves class stability, measured by the distance to the decision boundary, thereby leading to better generalization performance.

**Strengths:**

•	The work presents strong novelty and theoretical contributions.

•	The arguments are well-founded and clearly positioned within the existing literature.

**Weaknesses:**

•	Since the abstract track does not permit supplementary material, the submission lacks self-containment; several key results, lemmas, and theorems appear only in the appendix.

•	Figure 1 is not explained, which limits the reader’s understanding of its relevance to the main claims.

---

### Official Review · Reviewer_TUzi · 2025-11-01

**Soundness:** 4
**Correctness:** 3
**Rating:** 5
**Confidence:** 4

**Summary:**

The abstract presents a theoretic framework showing that in order for a classification model to achieve both low error and high stability or co-stability, the model has to be overparameterized, meaning that the amount of parameters needs to be roughly equal to the amount of datapoints times the desired distance for the stability. This implies that overparameterisation is necessary for robust generalization. The theoretical results are supported by some empirical evidence on neural networks trained on MNIST and CIFAR.

The problem is introduced in the context of understanding the behaviour of overparameterised neural networks in a statistical framework, where several challenges remain. The term "stability" is defined based on previous work, which is based on a distance metric of a datapoint, defined as the closest datapoint that is classified differently. This is generalized to "co-stability" which also captures infinite function classes. With some assumptions about the input distribution and function-set, a result about the Rademacher bound is stated. This is then used for the main result, which shows a relationship between the error, stability and overparameterization of a model with a high probability.

**Strengths:**

The results in the abstract provide progress towards understanding the theoretical aspects of neural networks, where several challenges remain open. This both helps bridge gaps in the statistical framework and can also potentially make it easier to train better performing models in practice. Specifically, the abstract shows a theoretical relationship between the error, stability and overparameterisation of a model, which implies that overparameterisation is necessary for robust generalization. This results should be of clear interest to the machine learning community. There are also some empirical experiments that confirm the claim from the theoretical statement.

As this contribution is presented in an abstract, the reviewer has not confirmed that the proofs in the appendix are sound or correct. However, the contributions should be of interest to the machine learning community anyhow.

**Weaknesses:**

All of the weaknesses below are well within the scope of expected shortcomings of an abstract, but some of the points may help the authors improve the final result.

- Figure 1:
  - The figure uses two different metrics in the $y$-axis. This is considered by many as an antipattern and should probably be avoided to reduce confusion.
  - The figure shows a relationship between accuracy (related to the error of the model) and the stability given an amount of parameters, but it is not sufficiently clear how this relates to the main result of the abstract. The ``Law of Robustness’’ states a relationship between stability, error and overparameterisation, and it therefore seems like it should be possible to tweak this relationship in different ways. Should one not be able to train highly accurate, but unstable, models with few parameters, and the other way around? From the main result it seems like the most natural way of confirming this empirically would be to show how different configurations impact each other, which would need more settings than just accuracy and stability being closely related to each other.
  - The figure shows some kind of measurement of the variance with the colored area around the lines, but it is not stated what this exactly is. Is it confidence intervals over multiple runs? What are the percentiles?
- While empirical results on MNIST and CIFAR strengthen the theoretical results, the authors should consider including more recent and difficult datasets. MNIST and CIFAR have both been incredibly important in the machine learning community, but in 2025 (and since about 2015) they are both considered extremely easy. If there is a reason that only very simple datasets like these can be used, the reason why should be stated.
- It is not clear what is meant with a "high probability" in line 121, and this is crucial for the actual result.
- There is an extra whitespace after the first parenthesis in line 053.

---

### Official Review · Reviewer_QCFv · 2025-11-03

**Soundness:** 3
**Correctness:** 2
**Rating:** 4
**Confidence:** 3

**Summary:**

The abstract attempts to establish a relationship between robustness, generalization and overparameterization in discontinuous classifiers and suggesting that overparameterization may be necessary for achieving both robustness and generalization with empirical results indicating that stability grows with model size.

**Strengths:**

- The abstract aims to offer a valuable and novel theoretical contribution to understanding the relationship between robustness, generalization, and overparameterization.

**Weaknesses:**

- The abstract is dense and jargon-heavy, introducing multiple theoretical terms without clear definitions or context, which makes it difficult to follow and evaluate.
- The abstract describes theoretical results and empirical findings, but it provides very little detail on the experimental methodology.

---

### Decision · Program_Chairs · 2025-11-05

**Decision:**

Accept

**Comment:**

The abstract is of interest to the community and should be presented at the conference.